# Risk of cancer in patients with insomnia: Nationwide retrospective cohort study (2009–2018)

Kichul Yoon[1], Cheol Min Shin[2]*, Kyungdo Han[3]*, Jin Hyung Jung[4], Eun Hyo Jin[5], Joo Hyun Lim[5], Seung Joo Kang[5], Yoon Jin Choi[6], Dong Ho Lee[2]

1 Department of Internal Medicine, Seonam Hospital, Seoul, South Korea, 2 Department of Internal Medicine, Seoul National University Bundang Hospital, Seongnam, South Korea, 3 Department of Statistics and Actuarial Science, Soongsil University, Seoul, South Korea, 4 Department of Medical Statistics, College of Medicine, The Catholic University of Korea, Seoul, South Korea, 5 Department of Internal Medicine, Healthcare Research Institute, Healthcare System Gangnam Center, Seoul National University Hospital, Seoul, South Korea, 6 Division of Gastroenterology, Department of Internal Medicine, Yonsei University College of Medicine, Seoul, South Korea

* scm6md@gmail.com (CMS); hkd917@naver.com (KH)

**Data Availability Statement:** All relevant data are within the manuscript and its Supporting information files.

**Funding:** This research was supported by a grant of the Korea Health Technology R&D Project

## Abstract

### Purpose

To investigate the association between insomnia and the risk of various cancers using the Korean National Health Insurance Service database.

### Materials and methods

Patients who underwent a national health examination in 2009 were followed-up until 2018. Newly-diagnosed cancers were collected one year after the baseline. Insomnia was defined as having a diagnosis of F510 or G470 within one year prior to enrollment. The incidence of various cancers was compared between patients with and without insomnia.

### Results

In the overall study population (N = 3,982,012), the risk for any type of cancer was not different between controls and insomnia patients (adjusted hazard ratio [aHR]: 0.990). However, it was different by age; insomnia increased the risk of any cancer in younger age groups (20–39y and 40–59y, aHR:1.310 and 1.139, respectively) but it significantly decreased the risk in the 60–79y age group (aHR: 0.939). In cancer type, colorectal cancer risk was lower (aHR: 0.872, *P* < 0.0001), whereas leukemia risk was higher (aHR: 1.402, *P* < 0.0001) in patients with insomnia than in those without it, regardless of sex. In men, the risk of stomach cancer was lower (aHR: 0.882, *P* = 0.0003), and the risks of lung (aHR:1.114, *P* = 0.0005), kidney (aHR 1.226, *P* = 0.0107), and prostate (aHR:1.101, *P* = 0.0028) cancers were higher in insomnia patients than in control patients. In women, insomnia patients compared to control patients showed a lower risk of ovarian cancer (aHR:0.856, *P* = 0.0344, respectively), while they had a higher risk of oral (aHR:1.616, *P* = 0.002), thyroid (aHR:1.072, *P* = 0.0192), and nerve (aHR: 1.251, *P* = 0.016) cancers.

through the Korea Health Industry Development Institute (KHIDI), funded by the Ministry of Health & Welfare, Republic of Korea (grant number: HI18C1140). The funders had no role in study design, data collection and analysis, decision to publish, or preparation of the manuscript.

**Competing interests:** The authors have declared that no competing interests exist.

## Conclusion

Insomnia is associated with an increased or decreased risk of some cancers, depending on age, cancer type and sex.

## Introduction

Insomnia has been widely investigated in terms of mental health, cardiovascular disease, and cancer [1–3]. Many factors, including demographic background, psychological status, and underlying medical and social conditions, affect the etiology of insomnia [4]. In the United States, the prevalence of insomnia disorder was reported to be 10–22% [5, 6]. In 2013, the prevalence of insomnia was 7.20% in males and 4.32% in females in Korea [7]. Interestingly, a Korean study showed survival of insomnia patients was lower than that of people without insomnia [7]. The association between insomnia and increased risk of various health conditions, such as depression, dementia, nonalcoholic fatty liver disease, hypertension and cardiovascular disease, has been suggested [3].

Several epidemiological reports have described the association between insomnia and cancer [2]. A nationwide case-control study in Taiwan showed that patients with insomnia exhibited significantly increased risks for overall cancer (adjusted hazard ratio [aHR]: 1.71, 95% confidence interval [CI]: 1.66–1.77) and many types of cancers (tracheal, nasal, liver, cervical, oral, colon, lymphatic, thyroid, myeloma, prostate, bladder, and kidney) [8]. A recent meta-analysis showed that there was a 24% overall increase of cancer risk in insomnia patients [2]. However, according to a systemic review and metaanalysis of previous epidemiologic studies, the results were inconsistent in terms of the risk for any kind of cancers and insomnia. Four of the 8 epidemiologic studies showed no relationship between insomnia and the risk of cancer. In contrast, four other reports showed a higher risk of cancer among study participants who suffered from insomnia [2]. In particular, breast cancer showed positive association between insomnia and cancer in two studies [9, 10]. In contrast, another study with more specific analyses of sleep pattern reported most sleep characteristics including hours of sleep per night did not correlate with breast cancer risk [11]. The metaanalysis also reported sex differences, showing that the association was significant only in women, but not in men [2]. However, the meta-analysis only covered limited types of cancers. The possible influence of insomnia on cancer development has been suggested based on several biological pathways, including melatonin, circadian clock gene dysregulation, and appetite-regulating hormones (leptin and ghrelin) [2]. However, insomnia may affect the patient's psychosocial factors and/or medical utilization behavior to modify the risk for cancers.

The purpose of this nationwide, retrospective cohort study of large scale, compared with previous reports, was to explore the correlation between various types of cancer and insomnia, based on information provided by the Korea National Health Insurance Service (KNHIS) database.

## Materials and methods

### Study setting and database

Data from KNHIS were used for the analysis. Individuals enrolled in the KNHIS who were ≥ 20 years old underwent standardized health examinations every two years. Height, weight, waist circumference, body mass index (BMI), blood pressure, lifestyle questionnaires

(smoking, alcohol, physical activities, etc.), laboratory results (aspartate aminotransferase (AST), alanine aminotransferase (ALT), gamma-glutamyl transferase(GGT), fasting glucose, creatinine, hemoglobin, estimated glomerular filtration rate (e-GFR), triglyceride (TG), and low-density lipoprotein (LDL) were screened.

## Study population

We first included all subjects who underwent a national health examination in 2009, and 40% of them were randomly selected in this analysis. Due to the regulations of the Korean National Health Insurance Sharing Services (NHISS) which limit total sample number and data size, we could only get a 40% simple random sampling database of the subjects who underwent a health-screening examination in 2009. Also, there is a limitation to the access of the KNHIS database due to institutional regulation. That is, we need approval from the institution to get access to the closed server, and we also need to make a reservation to use the database for a limited time. Patients with any cancer diagnosis prior to 2009 were excluded from the study. Patients newly diagnosed with cancer one year after the examination were excluded. Subjects with missing baseline characteristics were excluded from the study. We then collected the follow-up data of the baseline year 2009-health examinee until 2018 for any new cancer diagnosis.

## Ethical statement

This study was approved by the Institutional Review Board (IRB) of Seoul National University Bundang Hospital (IRB No.: X-1608/360-904). Since the study was retrospective and excluded any information on personal identification, the IRB waived the requirement for informed consent.

## Data collection and design

Baseline characteristics were collected on the date of the health-screening examination. A standardized self-report questionnaire was obtained at the time of the health check-up examination. Age, sex, income level (low income, defined as the lowest quintile among people under the KNHIS and Medical Aid recipients), smoking status, and alcohol consumption were included. BMI was classified into five levels as low ($< 18.5$ kg/m$^2$), normal (18.5–22.9 kg/m$^2$), overweight (23–24.9 kg/m$^2$), obese (25–29.9 kg/m$^2$), or severely obese ($\geq 30$ kg/m$^2$). The eGFR was calculated from the serum creatinine level. Fasting glucose, total cholesterol, and TG and LDL levels were measured.

Disease diagnoses in the database were classified according to the International Classification of Diseases Tenth Revision-Clinical Modification (ICD-10-CM) codes. To diagnose insomnia, ICD-10 requires at least 1 month of symptoms not explained by other specific causes [4]. We defined insomnia as having the diagnosis code F510 (nonorganic insomnia) or G470 (insomnia) within one year of 2009. The prevalence of insomnia in the study population is presented as S1 Table. All participants meeting the inclusion criteria were classified according to the presence of an insomnia diagnosis. We collected newly diagnosed cancer code data from the subjects until 2018 using the National Health Insurance database. We further classified insomnia diagnosis into 'existing' and 'newly diagnosed' insomnia, defining existing insomnia as already having insomnia diagnosis 5, 3, or 1 year(s), respectively, before the health checkup.

A cancer diagnosis was defined when both ICD-10 codes and cancer-specific insurance claim codes (V193 code) to receive national financial support were present. The cancer types were classified as follows: oral cancer (C00–13), esophageal cancer (C15), colorectal cancer

(C18 to C20), stomach cancer (C16), liver cancer (C22), pancreatic cancer (C25), lung cancer (C33,34), thyroid cancer (C73), gallbladder cancer (C23), biliary cancer (C24), laryngeal cancer (C32), renal cancer (C64), bladder cancer (C67), nerve cancer (C70 to C72), non-Hodgkin lymphoma (C82–C86, and C96), multiple myeloma (C90), leukemia (C91 to 95), skin cancer (C43, C44), prostate cancer (C61), testicular cancer (C62), breast cancer (C50), cervical cancer (C53), corpus cancer (C54), and ovarian cancer (C56).

## Statistical analysis

Continuous variables are presented as mean ± standard deviation (SD). Categorical variables are presented as frequencies and percentages. Student's t-test was performed for the comparison of continuous variables and the $\chi^2$-test was used for categorical variables [12, 13]. Cancer incidence was determined by dividing the number of events by the number of person-years. The Cox proportional hazards model was used after stratifying for covariates, including age, sex, low income, smoking, alcohol consumption, diabetes, hypertension, dyslipidemia, and BMI. The results were presented as adjusted hazard ratios (aHR) with 95% confidence intervals (CI). All statistical analyses were performed using SAS version 9.4 (SAS Institute, Cary, NC, USA) and R version 3.2.3 (The R Foundation for Statistical Computing, Vienna, Austria).

## Results

### Baseline characteristics

Among who underwent health-screening examinations, 40% of them were randomly selected in this analysis and a total of 4,234,339 people were enrolled in this study. After excluding those who already had cancer diagnoses at the time of health check-up and those with missing values, we also excluded subjects with newly diagnosed cancer within one year of screening examination. The enrollment flowchart is shown in Fig 1.

After exclusion, 3,982,012 participants were included in the analysis. The mean age of the subjects without cancer throughout the follow-up period was 48.28 and 56.77 in those with a cancer diagnosis.

### Comparison between control and newly diagnosed cancer

There were no significant differences in sex between the cancer and non-cancer groups (Table 1). In contrast, age distribution, smoking, alcohol consumption, and BMI were significantly different between the control and cancer patients. Waist circumference was significantly higher in patients with cancer. Cancer patients showed a significantly higher percentage in terms of low income, insomnia, diabetes, hypertension, and dyslipidemia than non-cancer patients. (Table 1.)

### Hazard ratio of the cancer incidence group and control regarding sleep disorder

In the overall study population, the risk for any kind of cancer was not different between control and insomnia groups (aHR, 0.990; 95% CI: 0.971–1.010, Table 2). However, the risk for colorectal cancer was significantly lower (aHR, 0.872; 95% CI: 0.833–0.914, $P < 0.0001$), while HR for leukemia was significantly higher (aHR: 1.402, 95% CI: 1.219–1.613, $P < 0.0001$) in insomnia patients than in controls, regardless of sex.

Among men, the risk of stomach cancer was significantly lower (aHR, 0.882; 95% CI: 0.825–0.944, $P = 0.0003$) in insomnia patients than in controls. In contrast, the risk of lung cancer (aHR, 1.114; 95% CI: 1.048–1.184, $P = 0.0005$), renal cancer (aHR: 1.226, 95% CI:

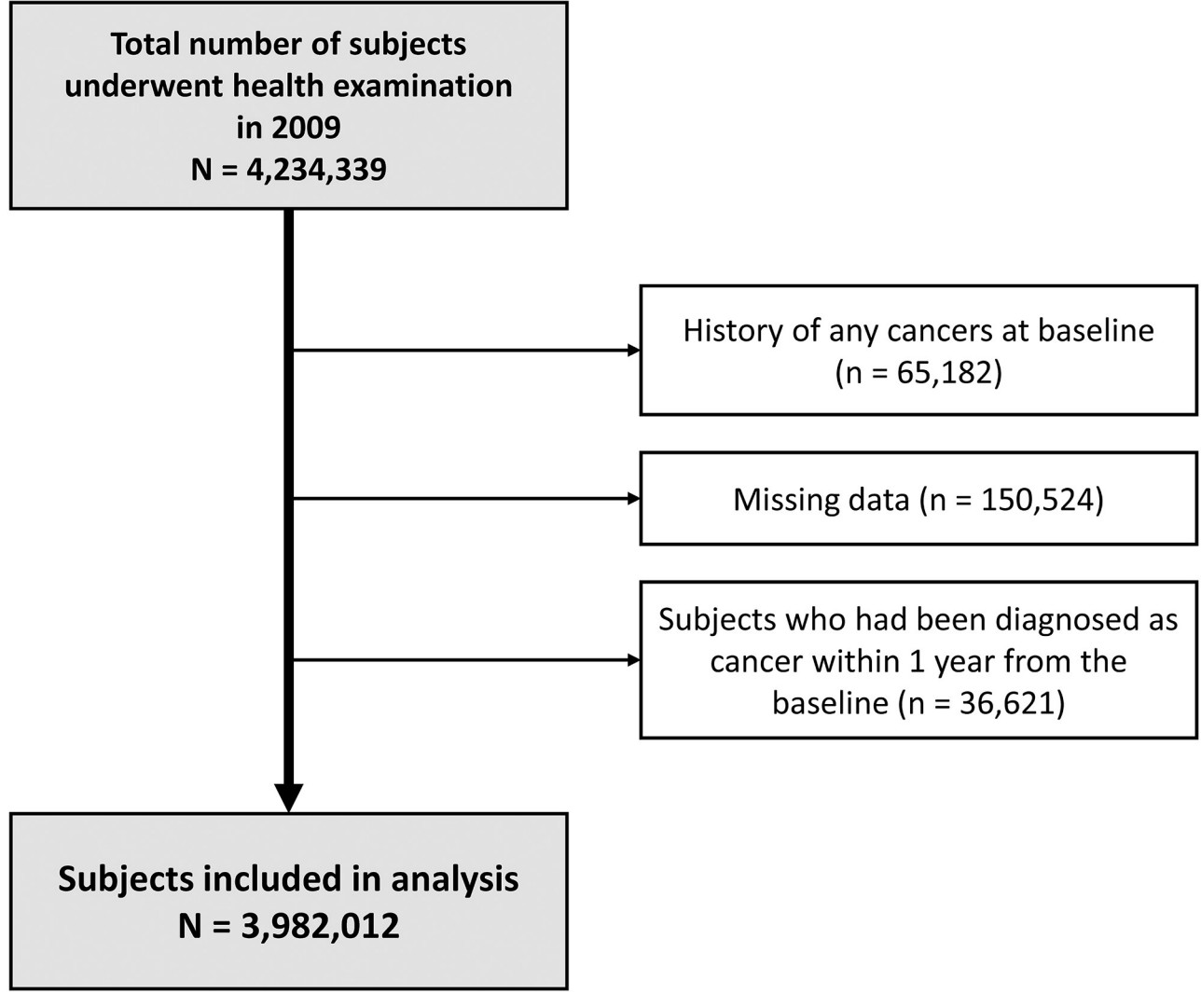

**Fig 1. Study flow chart showing the enrollment of study population.**

1.048–1.434, *P* = 0.0107), and prostate cancer (aHR: 1.101, 95% CI: 1.034–1.172, *P* = 0.0028) was higher in male insomnia patients (Table 3).

Among women, the risk of oral cancer (aHR, 1.616; 95% CI: 1.191–2.192, *P* = 0.002), thyroid cancer (aHR: 1.072, 95% CI: 1.011–1.136, *P* = 0.0192), and nerve cancer (aHR: 1.251, 95% CI: 1.043–1.500, *P* = 0.016) showed a significantly higher hazard ratio in insomnia patients than in controls. However, the risk of ovarian cancer significantly decreased in insomnia patients (aHR, 0.856; 95% CI: 0.741–0.989, *P* = 0.0344; Table 3).

## Subgroup analysis by age strata

We performed a subgroup analysis by age groups (20–39y, 40–59y, 60–79y, and ≥ 80y). The HR of cancer in insomnia patients of each age group was obtained, and *P*-values for interaction was calculated (Fig 2 and S2 Table). Fig 2 shows the results of cancers with a *P* for interaction < 0.05. In overall cancer, *younger* age groups (20–39y and 40–59y) showed significantly higher risk of cancers in patients with insomnia than in controls (aHR [95% CI], 1.310

**Table 1. Baseline characteristics of cancer and non-cancer subjects at baseline.**

| | Controls | Cancer patients | P-value |
|---|---|---|---|
| | (n = 3,772,596) | (n = 209,416) | |
| Age | 46.28 ± 13.90 | 56.77 ± 12.95 | < .0001 |
| < 40 | 1,258,821 (33.37) | 19,793 (9.45) | < .0001 |
| 40–64 | 2,077,191 (55.06) | 124,599 (59.50) | |
| ≥ 65 | 436,584 (11.57) | 65,024 (31.05) | |
| Sex | | | 0.4082 |
| Male | 2,083,086 (55.22) | 115,825 (55.31) | |
| Female | 1,689,510 (44.78) | 93,591 (44.69) | |
| Smoking | | | < .0001 |
| Non | 2,229,772 (59.10) | 125,295 (59.83) | |
| Ex | 534,382 (14.16) | 34,836 (16.63) | |
| Current | 1,008,442 (26.73) | 49,285 (23.53) | |
| Drinking | | | < .0001 |
| Non | 1,911,158 (50.66) | 120,362 (57.48) | |
| Mild[1] | 1,559,192 (41.33) | 71,585 (34.18) | |
| Heavy[2] | 302,246 (8.01) | 17,469 (8.34) | |
| Body mass index (kg/m$^2$) | 23.70 ± 3.47 | 23.97 ± 3.16 | < .0001 |
| <18.5 | 140,616 (3.73) | 5,860 (2.80) | < .0001 |
| 18.5–22.9 | 1,474,184(39.08) | 75,250 (35.93) | |
| 23–24.9 | 927,373 (24.58) | 53,920 (25.75) | |
| 25–29.9 | 1,096,092(29.05) | 66,669(31.84) | |
| ≥ 30 | 134,331(3.56) | 7,717 (3.69) | |
| Low income[3] | 658,937 (17.47) | 38,153 (18.22) | < .0001 |
| Waist circumference (cm) | 80.14 ± 9.50 | 82.03 ± 9.17 | < .0001 |
| Comorbidities | | | |
| Diabetes | 312,537 (8.28) | 30,738 (14.68) | < .0001 |
| Hypertension | 975,384 (25.85) | 87,010 (41.55) | < .0001 |
| Dyslipidemia | 673,644 (17.86) | 47,417 (22.64) | < .0001 |
| Insomnia | 123,319 (3.27) | 10,749 (5.13) | < .0001 |
| Laboratory findings | | | |
| Fasting serum glucose (mg/dL) | 96.96 ± 23.62 | 101.34 ± 27.51 | < .0001 |
| Total cholesterol (mg/dL) | 195.30 ± 41.56 | 195.57 ± 41.67 | 0.0042 |
| HDL-C (mg/dL) | 56.53 ± 32.96 | 55.75 ± 34.90 | < .0001 |
| LDL-C (mg/dL) | 121.4 ± 217.97 | 118.04 ± 141.47 | < .0001 |
| GFR (mL/min/1.73m$^2$) | 87.77 ± 45.56 | 84.66 ± 37.68 | < .0001 |
| Triglyceride (mg/dL)[4] | 112.54 (112.47–112.61) | 116.98 (116.70–117.26) | < .0001 |

Values are presented as mean ± standard deviation or numbers (percentage).

[1]Alcohol consumption < 30 g/day.

[2]Alcohol consumption ≥ 30 g/day.

[3]The lowest quintile range of yearly income.

[4]Geometric mean (95% confidence interval)

[1.154–1.487] and 1.139 [1.110–1.180], respectively). In the 60–79y age group, however, we found a s lower aHR for overall cancers in insomnia patients (aHR, 0.939; 95% CI: 0.916–0.963). Colorectal, liver, lung, thyroid, bladder, prostate, and breast cancers showed significant interaction of age with the association between insomnia and the cancer risks (*P* for

**Table 2. Risk of cancers according to insomnia in the overall study population.**

| | Sleep Disorder | n | Cancer Event | Follow-up Duration (person-years) | Crude incident rates per 1000 | aHR (95% CI)[1] | P-value |
|---|---|---|---|---|---|---|---|
| All cancer | No | 3847944 | 198667 | 31067930.8 | 6.3946 | 1(Ref.) | 0.3146 |
| | Yes | 134068 | 10749 | 1039129.05 | 10.3442 | 0.990(0.971,1.010) | |
| Stomach | No | 3847944 | 31392 | 31615581.3 | 0.99293 | **1(Ref.)** | **0.0006** |
| | Yes | 134068 | 1590 | 1067543.69 | 1.4894 | **0.915(0.870,0.963)** | |
| Colorectal | No | 3847944 | 37595 | 31598844.4 | 1.18976 | **1(Ref.)** | **< .0001** |
| | Yes | 134068 | 1921 | 1066755.38 | 1.80079 | **0.872(0.833,0.914)** | |
| Liver | No | 3847944 | 13957 | 31690559.3 | 0.44042 | 1(Ref.) | 0.9957 |
| | Yes | 134068 | 709 | 1071574.67 | 0.66164 | 1.000(0.927,1.080) | |
| Pancreatic | No | 3847944 | 15267 | 31692152.1 | 0.48173 | 1(Ref.) | 0.7077 |
| | Yes | 134068 | 947 | 1071356.8 | 0.88393 | 1.013(0.948,1.082) | |
| Lung | No | 3847944 | 25265 | 31677513.6 | 0.79757 | **1(Ref.)** | **0.0122** |
| | Yes | 134068 | 1723 | 1070114.95 | 1.61011 | **1.065(1.014,1.119)** | |
| Thyroid | No | 3847944 | 32551 | 31573104.6 | 1.03097 | **1(Ref.)** | **0.0403** |
| | Yes | 134068 | 1397 | 1066202.87 | 1.31026 | **1.058(1.003,1.117)** | |
| Lymphoma | No | 3847944 | 4660 | 31710782.5 | 0.14695 | 1(Ref.) | 0.8275 |
| | Yes | 134068 | 244 | 1072490.9 | 0.22751 | 1.015(0.891,1.156) | |
| Oral | No | 3847944 | 1224 | 31720569.9 | 0.038587 | 1(Ref.) | 0.4552 |
| | Yes | 134068 | 72 | 1072824.21 | 0.067113 | 1.096(0.861,1.395) | |
| Esophagus | No | 3847944 | 2511 | 31718352.9 | 0.07917 | 1(Ref.) | 0.8495 |
| | Yes | 134068 | 142 | 1072733.28 | 0.13237 | 1.017(0.857,1.206) | |
| Gallbladder | No | 3847944 | 2794 | 31719070.4 | 0.08809 | 1(Ref.) | 0.4312 |
| | Yes | 134068 | 219 | 1072689.74 | 0.20416 | 1.057(0.920,1.215) | |
| Biliary | No | 3847944 | 7326 | 31712641.8 | 0.23101 | 1(Ref.) | 0.7782 |
| | Yes | 134068 | 509 | 1072308.97 | 0.47468 | 1.013(0.925,1.109) | |
| Laryngeal | No | 3847944 | 1322 | 31719797.7 | 0.041677 | 1(Ref.) | 0.4581 |
| | Yes | 134068 | 78 | 1072790.88 | 0.072708 | 1.091(0.866,1.375) | |
| Kidney | No | 3847944 | 5091 | 31707131.1 | 0.16056 | 1(Ref.) | 0.0550 |
| | Yes | 134068 | 282 | 1072123.21 | 0.26303 | 1.126(0.997,1.272) | |
| Bladder | No | 3847944 | 6076 | 31705022.2 | 0.19164 | 1(Ref.) | 0.4586 |
| | Yes | 134068 | 385 | 1071877.87 | 0.35918 | 1.04(0.937,1.154) | |
| Nerves | No | 3847944 | 3290 | 31716886.4 | 0.10373 | **1(Ref.)** | **0.0315** |
| | Yes | 134068 | 210 | 1072572.49 | 0.19579 | **1.168(1.014,1.346)** | |
| Multiple myeloma | No | 3847944 | 2951 | 31716355.2 | 0.09304 | 1(Ref.) | 0.6491 |
| | Yes | 134068 | 172 | 1072661.99 | 0.16035 | 1.037(0.887,1.211) | |
| Leukemia | No | 3847944 | 2992 | 31717768.1 | 0.09433 | **1(Ref.)** | **< .0001** |
| | Yes | 134068 | 217 | 1072618.05 | 0.20231 | **1.402(1.219,1.613)** | |
| Skin | No | 3847944 | 5535 | 31706196.2 | 0.17457 | 1(Ref.) | 0.7077 |
| | Yes | 134068 | 408 | 1071699.84 | 0.3807 | 1.020(0.921,1.129) | |

[1]Adjusted for age, sex, low income, smoking, alcohol consumption, diabetes, hypertension, dyslipidemia and body mass index. aHR, adjusted hazard ratio; CI, confidence interval. Bold style indicates statistical significance.

interaction < 0.05). Ovarian cancer showed higher aHR in the age 20–39y group (aHR 1.854, 95% CI: 1.110–3.096). In the 40–59y age group, many cancers such as liver, pancreatic, oral, esophageal, gallbladder, renal, bladder, and male prostate cancer showed higher risk in insomnia patients than in controls (S2 Table). In the 60–79y age group, nerve cancer, multiple myeloma, and leukemia showed a higher risk (aHR [95% CI], 1.231 [1.041–1.455], 1.242

**Table 3. Risk of cancers according to insomnia stratified by sex.**

| | Male | | | | | | | Female | | | | | | |
|---|---|---|---|---|---|---|---|---|---|---|---|---|---|---|
| | Sleep Disorder | n | Cancer Event | Follow-up Duration (person-years) | IR per 1000 | aHR(95% CI)[1] | P-value | Sleep Disorder | n | Cancer Event | Follow-up Duration (person-years) | IR per 1000 | aHR(95% CI)[1] | P-value |
| All cancer | No | 2150819 | 110798 | 17325526.1 | 6.3951 | 1(Ref.) | 0.3922 | No | 1697125 | 87869 | 13742404.7 | 6.394 | 1(Ref.) | 0.4465 |
| | Yes | 48092 | 5027 | 358380.48 | 14.027 | 1.013 (0.984,1.042) | | Yes | 85976 | 5722 | 680748.58 | 8.40545 | 1.011 (0.984,1.038) | |
| Stomach | No | 2150819 | 22614 | 17581965.8 | 1.2862 | 1(Ref.) | **0.0003** | No | 1697125 | 8778 | 14033615.6 | 0.6255 | 1(Ref.) | 0.7470 |
| | Yes | 48092 | 908 | 369449.19 | 2.45771 | **0.882 (0.825,0.944)** | | Yes | 85976 | 682 | 698094.5 | 0.97695 | 0.987 (0.912,1.068) | |
| Colorectal | No | 2150819 | 22774 | 17584412.4 | 1.29512 | 1(Ref.) | **<.0001** | No | 1697125 | 14821 | 14014432 | 1.05755 | 1(Ref.) | **0.0152** |
| | Yes | 48092 | 873 | 369601.45 | 2.362 | **0.842 (0.787,0.902)** | | Yes | 85976 | 1048 | 697153.93 | 1.50325 | **0.925 (0.868,0.985)** | |
| Lung | No | 2150819 | 18109 | 17629744.9 | 1.02718 | 1(Ref.) | **0.0005** | No | 1697125 | 7156 | 14047768.7 | 0.5094 | 1(Ref.) | 0.9889 |
| | Yes | 48092 | 1122 | 370855.33 | 3.02544 | **1.114 (1.048,1.184)** | | Yes | 85976 | 601 | 699259.62 | 0.85948 | 1.001 (0.920,1.088) | |
| Oral | No | 2150819 | 849 | 17657729.8 | 0.048081 | 1(Ref.) | 0.0676 | No | 1697125 | 375 | 14062840.1 | 0.026666 | 1(Ref.) | **0.0020** |
| | Yes | 48092 | 24 | 372427.98 | 0.064442 | 0.683 (0.454,1.028) | | Yes | 85976 | 48 | 700396.23 | 0.068533 | **1.616 (1.191,2.192)** | |
| Renal | No | 2150819 | 3644 | 17648250 | 0.20648 | 1(Ref.) | **0.0107** | No | 1697125 | 1447 | 14058881.2 | 0.10292 | 1(Ref.) | 0.7169 |
| | Yes | 48092 | 168 | 371954.45 | 0.45167 | **1.226 (1.048,1.434)** | | Yes | 85976 | 114 | 700168.76 | 0.16282 | 1.036 (0.855,1.257) | |
| Nerves | No | 2150819 | 1864 | 17656493.6 | 0.10557 | 1(Ref.) | 0.5379 | No | 1697125 | 1426 | 14060392.8 | 0.10142 | 1(Ref.) | **0.0160** |
| | Yes | 48092 | 80 | 372391.74 | 0.21483 | 1.074 (0.856,1.346) | | Yes | 85976 | 130 | 700180.75 | 0.18567 | **1.251 (1.043,1.500)** | |
| Leukemia | No | 2150819 | 1876 | 17656199.1 | 0.10625 | 1(Ref.) | **0.0025** | No | 1697125 | 1116 | 14061569.1 | 0.07937 | 1(Ref.) | **0.0001** |
| | Yes | 48092 | 100 | 372321.02 | 0.26859 | **1.370 (1.117,1.680)** | | Yes | 85976 | 117 | 700297.03 | 0.16707 | **1.461 (1.204,1.772)** | |
| Thyroid | No | 2150819 | 8494 | 17623482.3 | 0.48197 | 1(Ref.) | 0.7490 | No | 1697125 | 24057 | 13949622.3 | 1.72456 | 1(Ref.) | **0.0192** |
| | Yes | 48092 | 176 | 371706.89 | 0.47349 | 1.025 (0.882,1.191) | | Yes | 85976 | 1221 | 694495.98 | 1.75811 | **1.072 (1.011,1.136)** | |
| Prostate | No | 2150819 | 16850 | 17608741 | 0.95691 | 1(Ref.) | **0.0028** | - | - | - | - | - | - | - |
| | Yes | 48092 | 1046 | 369184.56 | 2.83327 | **1.101 (1.034,1.172)** | | - | - | - | - | - | - | |
| Ovarian | - | - | - | - | - | - | - | No | 1697125 | 4027 | 14052202.7 | 0.28657 | 1(Ref.) | **0.03344** |
| | - | - | - | - | - | - | | Yes | 85976 | 198 | 699995.09 | 0.28286 | **0.856 (0.741,0.989)** | |
| Breast | - | - | - | - | - | - | - | No | 1697125 | 17421 | 13998427.82 | 1.2445 | 1(Ref.) | 0.1356 |
| | - | - | - | - | - | - | | Yes | 85976 | 840 | 697277.74 | 1.20468 | 0.948 (0.884,1.017) | |

[1] Adjusted for age, low income, smoking, alcohol consumption, diabetes, hypertension, dyslipidemia and body mass index. aHR, adjusted hazard ratio; CI, confidence interval; IR, crude incidence rate. Bold style indicates statistical significance.

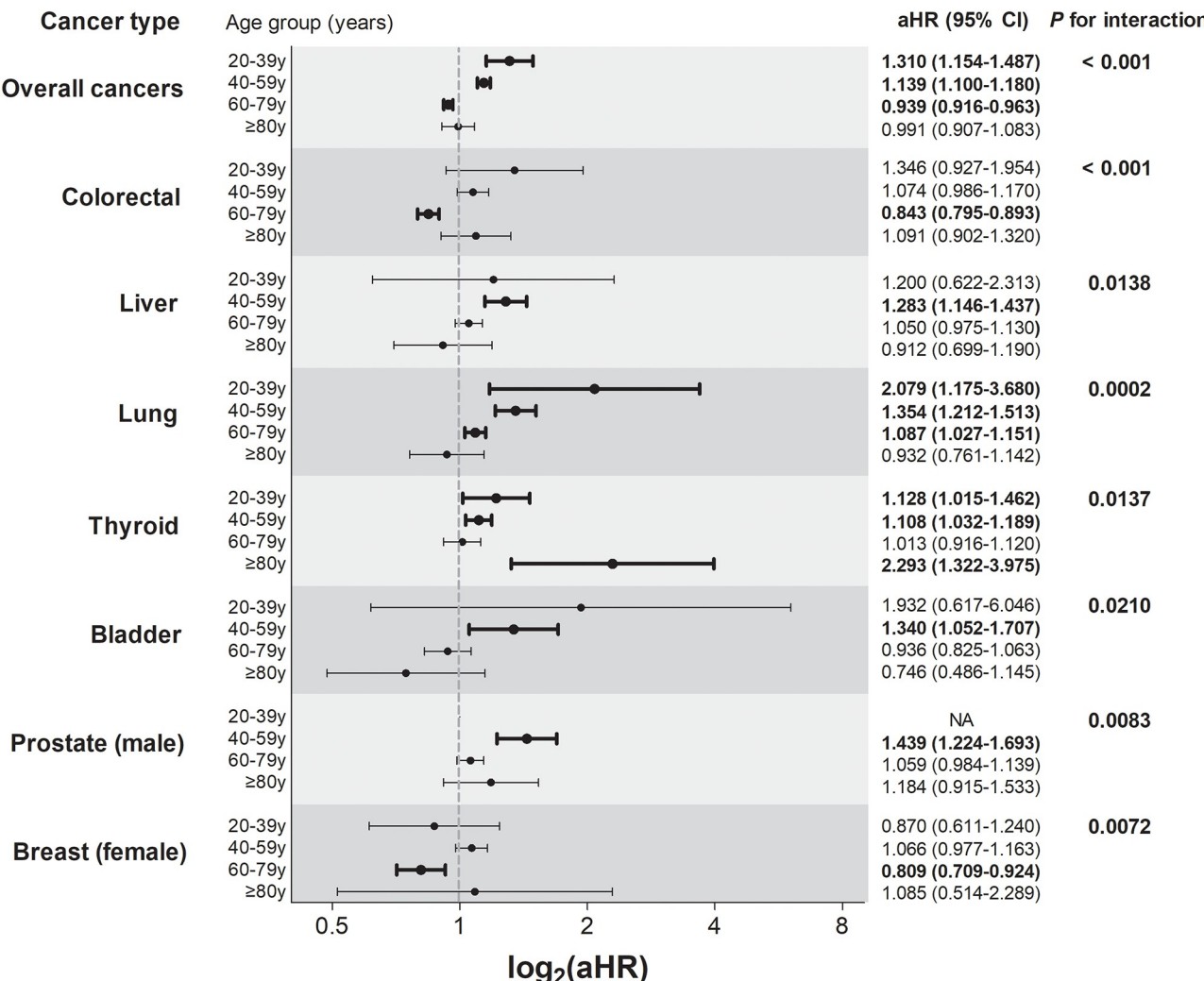

**Fig 2. Effect of insomnia on the risk of some cancers stratified by age group.** Results of all cancer types are presented in S2 Table. aHR, adjusted hazard ratio; CI, confidence interval. Thick lines and bold style indicate statistical significance ($p < 0.05$).

[1.017–1.516], 1.256 [1.051–1.503], respectively), while the risk of stomach, colorectal, and female breast cancers was lower in insomnia patients (aHR [95% CI], 0.903 [0.848–0.962], 0.843 [0.795–0.893] and 0.809 [0.709-.0.924], respectively.) (S2 Table).

## 'Existing' and 'newly diagnosed' insomnia

We further analyzed whether there was any difference in cancer incidence between 'existing' and 'newly diagnosed' insomnia. When the three different definitions (5,3,1 year(s)) were applied, there was no remarkable difference in terms of the aHR of each cancer incidence based on the insomnia diagnosis criteria (S3–S5 Tables).

## Discussion

There have been a few reports on the association between insomnia and cancer incidence; however, the results have been inconsistent [2, 3, 8]. In the present study, we found a difference in the association between insomnia and cancer risk according to age, sex and cancer site.

A previous study found that those with fewer than 6 hours of sleep per night had a markedly increased risk of colorectal adenomas compared with subjects that slept 7 hours or more [14], and a Taiwanese study found a higher HR of colon cancer in insomnia patients.(8) Another nested case-control study reported that the risk of colorectal cancer was higher in patients with insomnia [15]. Interestingly, in the current study, the risks of colon (both sexes), stomach (men), and ovarian cancers were decreased in patients with insomnia (Tables 2 and 3). This unexpected result might be attributed to the frequency of screening tests, such as endoscopy and gynecologic check-up, in insomnia patients, at least in part. Studies on the health behaviors of insomnia are scarce. Nevertheless, one study from the US found that insomnia was positively associated with uptake of colorectal cancer screening (odds ratio [OR]: 1.18, 95% CI: 1.06–1.32) in Caucasians [16]. If patients with insomnia receive colonoscopy more frequently, it may be possible to detect and remove colon adenomas in advance, thereby preventing the occurrence of colon cancer.

The lower risk of stomach cancer in men might also be attributed to their health checkup behavior. Therefore, studies on the association between insomnia and cancer should be interpreted in consideration of the level of medical care in the country or accessibility to cancer screening tests. In addition, the incidence of diffuse type gastric cancer, which we could not differentiate in the current data, is higher in young women than in men. As it is known that diffuse type gastric cancer is difficult to detect via screening endoscopy, it may have influenced the sex difference we found in our analysis [17]. To our knowledge, no researchers have reported a link between the important risk factor *Helicobacter pylori* status and insomnia. Other causes of stomach cancer, such as red meat or high salt intake, should also be considered [18, 19]. As an association between stress, dietary habits, and insomnia has been suggested, further investigation on dietary patterns of insomnia patients is needed to clarify this issue [20].

The risk of ovarian cancer differs according to factors such as parity status, lactation, and oral contraceptive pills [21]. We found a lower risk of ovarian cancer in patients with insomnia (Table 3). The result was not consistent with the previous Taiwanese case-control study, which reported a higher risk of the cancer among patients with insomnia [8]. However, one study in the US previously reported that menstrual cycle-related insomnia was associated with significantly decreased risk of ovarian cancer (OR: 0.5, 95% CI: 0.3–0.8) [22]. Associations with insomnia appear to differ by pathologic subtype [23]. Estrogen levels in female with insomnia might have affected the results, including our present study [24]. Results on ovarian cancer need to be stratified according to subtypes.

In contrast, a study on a Taiwanese sample reported most sleep characteristics did not correlate with breast cancer risk [11], which is consistent with our findings. Also, our finding of a higher risk of oral and prostate cancers is similar to the previous Taiwanese nationwide report [8]. However, we found a higher risk of oral cancer only in the female population. Risk factors for oral cancers include smoking and heavy alcohol drinking. Smoking and drinking are more prevalent in men, whereas these may play a minor role in female oral cancers. After adjusting for smoking and drinking, insomnia remains an independent risk factor for oral cancer in female patients but not in male patients. Thyroid cancer was reported to be significantly higher in insomnia patients in a meta-analysis [2]. We also reported a higher HR of thyroid cancer in female patients.

Reports on nervous system cancers and insomnia are scarce. However, a nationwide cohort study showed a significantly higher risk of primary CNS cancers in the obstructive sleep apnea subgroup [25]. We found that there was a higher risk of nervous system cancer in female patients. However, we could not independently assess this association as data for peripheral and nervous system cancers could not be collected separately.

A Finnish prospective population-based cohort of males showed that a sleep duration of less or more than 7–7.5 hours was associated with a higher lung cancer risk [26]. We also found a higher HR of lung cancer only in the male population. A recent study investigated the potential causality between genetically predicted insomnia and lung cancer risk. A Mendelian randomization analysis indicated an increased risk of lung cancer in insomnia patients [27].

Insomnia has been reported as a symptom of leukemia [28]. However, studies on the relationship between insomnia and leukemia are scarce. Insomnia, as a predisposing factor for leukemia, needs to be investigated.

In this study, we defined 'insomnia' as having a diagnosis within one year of 2009, at the time of the initial health check-up. Thus, we further tried to verify the association between insomnia and cancer risk by classifying the insomnia diagnosis into 'existing' and 'newly diagnosed' insomnia, defining 'existing' insomnia as already having insomnia diagnosis 5, 3, or 1 year(s) respectively before the health checkup (S3–S5 Tables). In this sensitivity analysis, we could not find any remarkable difference in results according to the diagnoses, considering up to 5 years of pre-existence of insomnia. Generally, as cancer development requires a substantial amount of time for multiple sequential mutations from the exposure of possible causes [29], the 5-year duration may not be long enough to evaluate the risk of cancer development. As insomnia is often thought to be under-diagnosed [30], and there could be substantial time between the onset of insomnia symptoms and medical help-seeking behavior for diagnosis, our cohort patients might have had insomnia for a longer time than previously reported.

Insomnia is closely related to major depressive disorders [31], which might also be associated with some cancers [32]. Coexistence of other psychiatric conditions should be considered. Furthermore, insomnia can be a clinical diagnosis, and it may also present as an epiphenomenon of other primary conditions [33]. Our result regarding unexpected lower risk of several cancers including colon (both sexes), stomach (men), and ovarian (female) malignancy might be attributed to insomnia associated with personality traits such as hypochondriasis or anxiety, and may inadvertently promote early detection of premalignant lesions by frequent endoscopic and gynecologic examination [34].

Age is an important confounding factor. Thus, we also performed subgroup analyses by age groups. In the overall study population, the risk for any type of cancer was not different between controls and insomnia patients. However, the risk of cancer in patients with insomnia revealed differences according to age group and the type of cancer. In younger age groups (20–39y and 40–59y), the risk of overall cancers was higher in patients with insomnia, whereas it was significantly lower in the 60–79y age group, especially in cases of colorectal, stomach, and female breast cancers. As colorectal cancer is a typical cancer that has a precancerous lesion and removal of adenomatous precursors can result in prevention of cancer progression in this age group [35], health behaviors could have affected our findings.

This study had several limitations. First, as mentioned above, this retrospective cohort study did not show a causal relationship between insomnia and cancer; thus, the interpretation of unexpectedly low hazard ratios of certain types of cancer is difficult. In particular, in colorectal cancer, further studies regarding colon adenoma in insomnia patients might show whether health-screening behavior could prevent the progression of adenoma to cancer in the Korean population. Second, we had limited use of the KNHIS claims database; thus, data on sedative-hypnotics and antidepressants could not be analyzed further. Additionally, the KNHIS claims database does not collect data on cancer stage or histology. Nevertheless, sex differences in the association between certain types of cancer and insomnia will be an interesting topic for future studies. In addition, reliance on ICD-10 codes and its inherent inaccuracies should be considered. In the Korean NHIS database, information on ICD-10 diagnosis should be provided by a physician who performs any diagnostic tests or procedures or

prescribes medications. Although information on the severity of insomnia is lacking, patients with the ICD-10 insomnia code might have insomnia symptoms severe enough to need medications. However, our findings should be confirmed in other ethnicities and countries.

In conclusion, insomnia may increase or decrease the risk of cancer, depending on age, cancer type and sex. Further studies are necessary to clarify the mechanisms underlying insomnia and cancer development.

## Supporting information

**S1 Table. Prevalence of ICD-10 code insomnia in this study population in the year 2009.**
(PDF)

**S2 Table. Risk of cancers according to insomnia stratified by age group.**
(PDF)

**S3 Table. Hazard ratio of cancer incidence according to 'preexisting' and 'newly diagnosed' insomnia, defining 'preexisting' as having insomnia diagnosis 5 years before 2009 health checkup.**
(PDF)

**S4 Table. Hazard ratio of cancer incidence according to 'preexisting' and 'newly diagnosed' insomnia, defining 'preexisting' as having insomnia diagnosis 3 years before 2009 health checkup.**
(PDF)

**S5 Table. Hazard ratio of cancer incidence according to 'preexisting' and 'newly diagnosed' insomnia, defining 'preexisting' as having insomnia diagnosis 1 year before 2009 health checkup.**
(PDF)

## Author Contributions

**Conceptualization:** Kichul Yoon, Cheol Min Shin, Kyungdo Han, Eun Hyo Jin, Dong Ho Lee.

**Data curation:** Kyungdo Han, Jin Hyung Jung.

**Formal analysis:** Jin Hyung Jung.

**Funding acquisition:** Cheol Min Shin.

**Investigation:** Kyungdo Han.

**Methodology:** Kyungdo Han.

**Project administration:** Cheol Min Shin, Kyungdo Han.

**Supervision:** Cheol Min Shin, Kyungdo Han, Dong Ho Lee.

**Writing – original draft:** Kichul Yoon.

**Writing – review & editing:** Kichul Yoon, Cheol Min Shin, Eun Hyo Jin, Joo Hyun Lim, Seung Joo Kang, Yoon Jin Choi, Dong Ho Lee.

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
