## [Decision Letter · Decision Letter 0]

2 Jan 2023

PONE-D-22-33830Risk of Cancer in Patients with Insomnia: Nationwide Retrospective Cohort Study (2009-2018)PLOS ONE

Dear Dr. Shin,

Thank you for submitting your manuscript to PLOS ONE. After careful consideration, we feel that it has merit but does not fully meet PLOS ONE’s publication criteria as it currently stands. Therefore, we invite you to submit a revised version of the manuscript that addresses the points raised during the review process.

We look forward to receiving your revised manuscript.

Kind regards,

Dong Keon Yon, MD, FACAAI

Academic Editor

PLOS ONE

Additional Editor Comments:

Thank you for submitting your manuscript. The reviewers and I believe it is of potential value for our readers. However, the reviewers have raised a number of very important issues, and their excellent comments will need to be adequately addressed in a revision before the acceptability of your manuscript for publication in the Journal can be determined.

Reviewers' comments:

Reviewer's Responses to Questions

**Comments to the Author**

1. Is the manuscript technically sound, and do the data support the conclusions?

Reviewer #1: Partly

Reviewer #2: Yes

Reviewer #3: Yes

2. Has the statistical analysis been performed appropriately and rigorously? 

Reviewer #1: Yes

Reviewer #2: Yes

Reviewer #3: Yes

3. Have the authors made all data underlying the findings in their manuscript fully available?

Reviewer #1: Yes

Reviewer #2: Yes

Reviewer #3: Yes

4. Is the manuscript presented in an intelligible fashion and written in standard English?

Reviewer #1: Yes

Reviewer #2: Yes

Reviewer #3: Yes

5. Review Comments to the Author

Reviewer #1: • This is a retrospective cohort study from the Korean National Health Insurance (KNHIS) database seeking to understand a potential link between the incidence of insomnia and risk of cancer diagnoses in this population.

• Adults (>=20 years old) enrolled in the KNHIS got standard exams every 2 years, which included lifestyle questionnaires and labs.

• Cohort selected 40% of subjects from 2009, followed over time until 2018. Previous cancer diagnoses excluded. Missing data excluded.

• Disease diagnoses classified by ICD-10 coding (including insomnia and cancers). Insomnia diagnoses stratified by incident or prevalent and timing. Cancer diagnoses included combined ICD-10 and insurance claim codes.

• Analyses for Hazard adjusted for age, sex, income, smoking, EtOH, Dm-2, HTN, HLD, BMI.

• 3.98M subjects included in analysis. Cancer diagnosis in older subjects (56.8 vs 48.3 years). No difference in sex between CA and non-CA diagnoses. CA diagnoses were higher waist circumference, metabolic syndrome (DM-2, HTN, HLD), lower income, higher insomnia, smoking, alcohol use, BMI, etc.

• Per specific cancers, insomnia group had higher rates of certain cancers (i.e. colorectal, leukemia), and lower in others (ovarian, stomach), which was different by gender as well. No difference in cancer diagnoses rates by type of insomnia (existing, newly diagnosed, etc).

• Summary – The authors present intriguing data regarding a potential association between a diagnosis of insomnia and the risk of developing cancer diagnoses. Strengths include a large and relatively comprehensive dataset utilizing the resources of a national healthcare system in Korea, and a robust analysis.

Comments

• The authors attempt to explain the lower rate of certain malignancies in insomnia patients (Discussion second paragraph), noting that insomniac patients may have higher rates of screening uptake for malignancy like colonoscopy. However, they go on to suggest the stomach cancer rate is potentially related to H. Pylori infection, however I fail to see how this is linked to insomnia, and I would remove that potential explanation (one could posit any number of comparable explanations, like they consume more food with nitrates that is linked to stomach cancer, but explanations to link insomnia to lower malignancy rates should require further justification).

• In the fourth discussion paragraph, the third sentence should probably indicate the authors explaining a link between insomnia and oral cancer in females (“…insomnia could independently be a risk factor for oral cancer in female patients”).

• In the fifth discussion paragraph, I would not report the hazard from the cited reference itself, the reference is already provided. Additionally, the following sentence (the third sentence) should read “…there was a higher risk of nervous system cancer in female patients”. The following sentence should indicate the cancers themselves as well, “However, we could not independently validate this association as we did not collect data on peripheral and nervous system cancers specifically”.

• In the eight discussion paragraph, discussing the stratification of insomnia and risk of cancer, the authors suggest the existing insomnia (within 5 years) is not long enough to fully evaluate the risk of cancer. This is a theory, not a fact, and should be treated as such, “…multiple sequential mutations from the exposure of possible causes (19), the 5-year duration may not be long enough to evaluate the risk of cancer development”.

• In the 9th discussion paragraph, the link between insomnia and lower cancer rates, secondary potentially to medical-care-seeking behavior, should be rephrased. (“…attributed to insomnia associated with personality traits such as hypochondriasis or anxiety, may inadvertently promote early detection of premalignant lesions…”).

• The authors fail to discuss or entertain a number of limitations in their study, namely the reliance on ICD-10 codes (and the inaccuracies therein), the lack of generalization outside a Korean population, the possibility of reporting bias (older subjects, who are more likely to develop cancer, may also be more likely to report insomnia, and age may be a significant confounder -see several references linking increased insomnia to advanced age, for example Miner B, et al. Sleep Med Clin 2017;12(1):31-38.), and how age may be a significant confounder in this analysis (the use of multivariable adjustment alone may not be sufficient to address this confounding).

• Why did the authors not perform a validation analysis on a subset of subjects from their database to ensure their findings could be replicated? They only used 40% of the initial dataset in the initial analysis, they should have data for a validation of a subset (i.e. 20% of the remaining dataset) to validate their initial findings. This is pertinent particularly given some of the unusual associations between insomnia and lower rates of malignancy they have observed.

• Age is a significant concern regarding confounding. The authors should consider performing a sub-analysis, where they stratify their data by age strata (i.e. every 5 or 10 year age groups as of 2009) and perform multivariable analyses on these subjects to assess insomnia with malignancy. This would better enable them to see if the association has the same value across age groups, They appear to have the dataset size to do this, and it would help alleviate concerns regarding the confounding nature of age in this longitudinal study. Understandably, older age groups would have less follow-up time, but additionally they would be more at risk of malignancy development, so I would be curious to see the results of this type of analysis, and if their findings of an association are maintained.

Reviewer #2: Insomnia is the most common sleep disorder and has been shown to be associated with an increased risk for numerous disorders, including cardiovascular, neurological and psychiatric disorders. As such, it merits particular attention in the research community. Yoon et al. describe the association between insomnia in various types of cancer in the Republic of Korea between 2009 and 2018. The authors show that insomnia is associated with an increased risk for several cancers, but also decreased risk for some types of cancer, including colorectal cancer. Due to the inconclusive nature of the connection between insomnia and various types of cancer, this study presents a valuable contribution to the existing literature and understanding behind this connection. While the article is easy to understand and fluent to read, there are several suggestions that, in my opinion, would make the article more comprehensive and novel, thus suitable for publication.

Abstract:

- Please write exact P values in the abstract where possible.

- Please be slightly more specific (most important/reliable findings) in the conclusion of the abstract.

Introduction:

- Please add very brief paragraph about the health implications of chronic insomnia after the first paragraph to fluently transition to the second paragraph about the association between insomnia and cancer.

- In the second paragraph, please add more background on why the results associating insomnia and cancer are inconsistent (and what cancers in particular) citing relevant literature.

- In the last paragraph, please clearly explain why your study is different compared to already published studies and how it may add value to the field (apart from being another group of patients).

Methods:

- Please explain the reasoning behind including 40% and not all of the patients.

- Please elaborate whether patients with all cancer stages were included and whether you grouped them in any way, which might add significant value to the literature. I would suggest grouping patients by either cancer stage (early/late…) and/or by age group/sex which would add another novelty factor to your research.

- Please explain the sleep history of these patients prior to inclusion if possible since presence/duration of insomnia/other sleep disorders might be an important factor in cancer development. I would suggest grouping patients by insomnia severity and duration to see whether there is any association with different types/stages of cancer, which would be another differentiating factor for your study.

- Please elaborate whether all patients had insomnia during the whole period (2009-2018), how their insomnia was followed and whether patients with cured insomnia between 2009 and 2018 were still included in the study till the very end.

Results:

- Please elaborate on why subjects with newly diagnosed cancer within one year of screening examination were excluded.

Discussion:

- Please elaborate in the discussion whether your findings could be influenced by the region you studied since some cancers are more/less common in the Asian population in general.

Figures:

- I suggest adding a figure (e.g. Forest plot) that will highlight the most important results in a visually appealing way, in addition to the tables where you can write all the results.

These suggestions are likely to make the article more comprehensive and novel and I feel they are necessary for this article to be accepted for publication.

Reviewer #3: Dear authors,

I have now completed the review of the manuscript titled "Risk of Cancer in Patients with Insomnia: Nationwide Retrospective Cohort Study (2009-2018)."

In the present study, the authors investigated the association between insomnia and the risk of various cancers using data from the Korean National Health Insurance Service database.

The manuscript is interesting and, in general, fair written.

I have some minor suggestions before recommending an "accept."

1. The background section introduced some relevant articles. Please summarize the results with effect sizes.

2. I suggest authors clarify how other researchers can obtain the original data.

3. How many people have F510 (nonorganic insomnia) or G470 (insomnia) each? Could you also compare the rate with previous studies?

4. In the ‘Statistical Analysis’ section, the paragraph is quite long but there are no references for selecting your method. Please add two references to the following sentence:

Student’s t-test was performed for the comparison of continuous variables and the χ2-test was used for categorical variables [1, 2].

[1] https://doi.org/10.54724/lc.2022.e1

[2] https://doi.org/10.54724/lc.2022.e3

5. What is the future scope of the proposed research, authors have described the limitations in a good way, and I suggest that these can be the future scope of the work.

6. PLOS authors have the option to publish the peer review history of their article (what does this mean?). If published, this will include your full peer review and any attached files.

Reviewer #1: **Yes: **Arun Jose

Reviewer #2: No

Reviewer #3: No

---

## [Author Response · Author response to Decision Letter 0]

4 Mar 2023

Please refer to the attached 'Respond to Reviewers' docx file. 

Thank you.

---

## [Decision Letter · Decision Letter 1]

17 Mar 2023

PONE-D-22-33830R1Risk of Cancer in Patients with Insomnia: Nationwide Retrospective Cohort Study (2009-2018)PLOS ONE

Dear Dr. Shin,

Thank you for submitting your manuscript to PLOS ONE. After careful consideration, we feel that it has merit but does not fully meet PLOS ONE’s publication criteria as it currently stands. Therefore, we invite you to submit a revised version of the manuscript that addresses the points raised during the review process.

We look forward to receiving your revised manuscript.

Kind regards,

Dong Keon Yon, MD, FACAAI

Academic Editor

PLOS ONE

Journal Requirements:

Additional Editor Comments :

Thank you for submitting your manuscript. The reviewers and I believe it is of potential value for our readers. However, the reviewers have raised a number of very important issues, and their excellent comments will need to be adequately addressed in a revision before the acceptability of your manuscript for publication in the Journal can be determined. We cannot guarantee that your revised paper will be chosen for publication; this would be solely based on how satisfactorily you have addressed the reviewer comments.

Reviewers' comments:

Reviewer's Responses to Questions

**Comments to the Author**

1. If the authors have adequately addressed your comments raised in a previous round of review and you feel that this manuscript is now acceptable for publication, you may indicate that here to bypass the “Comments to the Author” section, enter your conflict of interest statement in the “Confidential to Editor” section, and submit your "Accept" recommendation.

Reviewer #1: All comments have been addressed

Reviewer #2: All comments have been addressed

Reviewer #3: All comments have been addressed

2. Is the manuscript technically sound, and do the data support the conclusions?

Reviewer #1: Yes

Reviewer #2: Yes

Reviewer #3: Yes

3. Has the statistical analysis been performed appropriately and rigorously? 

Reviewer #1: Yes

Reviewer #2: Yes

Reviewer #3: Yes

4. Have the authors made all data underlying the findings in their manuscript fully available?

Reviewer #1: No

Reviewer #2: Yes

Reviewer #3: Yes

5. Is the manuscript presented in an intelligible fashion and written in standard English?

Reviewer #1: Yes

Reviewer #2: Yes

Reviewer #3: Yes

6. Review Comments to the Author

Reviewer #1: • The authors have done a good job of responding to my critiques. I find the stratification of insomnia by age to be quite interesting, significantly higher risk in the younger population, but curiously a decreased hazard in the holder age group, and the interaction between age, duration of insomnia, and other associated characteristics (health seeking behavior, hypochondriasis, etc.) is quite interesting.

• I would recommend the authors indicate the limitations of their database as it relates to why they had to use only 40% of the initial cohort, and also why it would be difficult for others to gain access to their data for validation. Per the authors, “according to institutional regulations, we could only get a 40% simple random sampling database of the subjects who underwent a health-screening examination in 2009. Also, there is a limitation to the access of the KNHIS database due to institutional regulation. That is, we need approval from the institution to get access to the closed server, and we also need to make a reservation to use the database for a limited time.”. I suggest paraphrasing this explanation go in the Methods – Study population section.

Reviewer #2: I would like to thank the authors for addressing my concerns and I think the article is ready to be accepted.

Reviewer #3: All comments have been addressed. Thank you to the authors and editors for considering my opinion on this manuscript.

7. PLOS authors have the option to publish the peer review history of their article (what does this mean?). If published, this will include your full peer review and any attached files.

Reviewer #1: **Yes: **Arun Jose

Reviewer #2: No

Reviewer #3: No

---

## [Author Response · Author response to Decision Letter 1]

22 Mar 2023

March 22nd, 2023

Dear Editor:

RE: "Risk of Cancer in Patients with Insomnia: Nationwide Retrospective Cohort Study (2009-2018)” (Manuscript ID: PONE-D-22-33830)

coauthored by Kichul Yoon, Jin Hyung Jung, Eun Hyo Jin, Joo Hyun Lim, Seung Joo Kang, Yoon Jin Choi, and Dong Ho Lee

Thank you very much for giving us one more opportunity for revision.

Accurate and kind comments by the reviewer have been addressed in the discussion. We also believe that these comments improved our manuscript. Changes have been made by changing the color to RED in the revised manuscript, tables and figure legends to avoid any confusion. 

I anticipate good response.

Thank you!

Sincerely,

Cheol Min Shin, M.D., Ph.D.

Kyungdo Han, Ph.D.

 

Reviewer #1

• The authors have done a good job of responding to my critiques. I find the stratification of insomnia by age to be quite interesting, significantly higher risk in the younger population, but curiously a decreased hazard in the holder age group, and the interaction between age, duration of insomnia, and other associated characteristics (health seeking behavior, hypochondriasis, etc.) is quite interesting.

• I would recommend the authors indicate the limitations of their database as it relates to why they had to use only 40% of the initial cohort, and also why it would be difficult for others to gain access to their data for validation. Per the authors, “according to institutional regulations, we could only get a 40% simple random sampling database of the subjects who underwent a health-screening examination in 2009. Also, there is a limitation to the access of the KNHIS database due to institutional regulation. That is, we need approval from the institution to get access to the closed server, and we also need to make a reservation to use the database for a limited time.”. I suggest paraphrasing this explanation go in the Methods – Study population section.

Answer: Thank you for your important comments that improved our manuscript. Following the reviewer’s recommendation, we revised regarding sentences in the Method - Study population section as follows (line 114, page 6); “… Due to the regulations of the Korean National Health Insurance Sharing Services (NHISS) which limit total sample number and data size, we could only get a 40 % simple random sampling database of the subjects who underwent a health-screening examination in 2009. Also, there is a limitation to the access of the KNHIS database due to institutional regulation. That is, we need approval from the institution to get access to the closed server, and we also need to make a reservation to use the database for a limited time. …” 

The authors really appreciated your kind and helpful comments. The revisions based on these comments improved the accuracy and the quality of our manuscript. Thank you again.

Kichul Yoon, M.D.

Cheol Min Shin, M.D., Ph.D.

 

Reviewer #2: I would like to thank the authors for addressing my concerns and I think the article is ready to be accepted.

Answer: Thank you for your important comments that improved our manuscript.

Reviewer #3: All comments have been addressed. Thank you to the authors and editors for considering my opinion on this manuscript.

Answer: Thank you for your important comments that improved our manuscript.

---

## [Decision Letter · Decision Letter 2]

3 Apr 2023

Risk of Cancer in Patients with Insomnia: Nationwide Retrospective Cohort Study (2009-2018)

PONE-D-22-33830R2

Dear Dr. Shin,

We’re pleased to inform you that your manuscript has been judged scientifically suitable for publication and will be formally accepted for publication once it meets all outstanding technical requirements.

Kind regards,

Dong Keon Yon, MD, FACAAI

Academic Editor

PLOS ONE

Additional Editor Comments (optional):

This is an excellent paper.

Reviewers' comments:

Reviewer's Responses to Questions

**Comments to the Author**

1. If the authors have adequately addressed your comments raised in a previous round of review and you feel that this manuscript is now acceptable for publication, you may indicate that here to bypass the “Comments to the Author” section, enter your conflict of interest statement in the “Confidential to Editor” section, and submit your "Accept" recommendation.

Reviewer #1: All comments have been addressed

2. Is the manuscript technically sound, and do the data support the conclusions?

Reviewer #1: (No Response)

3. Has the statistical analysis been performed appropriately and rigorously? 

Reviewer #1: (No Response)

4. Have the authors made all data underlying the findings in their manuscript fully available?

Reviewer #1: (No Response)

5. Is the manuscript presented in an intelligible fashion and written in standard English?

Reviewer #1: (No Response)

6. Review Comments to the Author

Reviewer #1: (No Response)

7. PLOS authors have the option to publish the peer review history of their article (what does this mean?). If published, this will include your full peer review and any attached files.

Reviewer #1: **Yes: **Arun Jose

---

## [Editor Report · Acceptance letter]

12 Apr 2023

PONE-D-22-33830R2 

Risk of Cancer in Patients with Insomnia: Nationwide Retrospective Cohort Study (2009-2018) 

Dear Dr. Shin:

I'm pleased to inform you that your manuscript has been deemed suitable for publication in PLOS ONE. Congratulations! Your manuscript is now with our production department. 

Kind regards, 

on behalf of

Dr. Dong Keon Yon 

Academic Editor

PLOS ONE